# Cable Eccentricity Detection Method Based on Magnetic Field

**DOI:** 10.3390/s24175525

**Published:** 2024-08-26

**Authors:** Yuandi Liu, Pengxuan Wei, Yinghong Zhang

**Affiliations:** School of Electromechanical Engineering, Guilin University of Electronic Technology, Guilin 541004, China; 13328280121@163.com (Y.L.); 17331972822@163.com (P.W.)

**Keywords:** eccentricity detection, magnetic field detection, law of electromagnetic induction, non-contact measurement

## Abstract

Amid the rapid advancement of electronic information technology, the need for cable eccentricity measurement in the industry is increasing both in China and across the globe. Current detection methods have several flaws, including high costs, insufficient accuracy, and instability. In this paper, we introduce a magnetic field-based detection method for cable eccentricity that provides high precision and cost-effectiveness. We position three pairs of magnetic field-collection modules in a circular array to gather magnetic flux density information induced by the electrified cable. We apply the law of electromagnetic induction to calculate the cable eccentricity. Our method is non-contact, preserving the cable’s integrity. Our method outperforms traditional detection methods, not only in achieving greater accuracy and stability but also in significantly lowering the detection cost. Simulations and experiments show that our method’s error rate under specified conditions is 0~4%, with a maximum standard deviation of 0.11, confirming its precision and stability in detecting cable eccentricity. The effectiveness of our method is influenced by two factors: lift-off value and loading current intensity. Our method presents a novel concept and a dependable strategy for the progress of cable eccentricity-detection technology.

## 1. Introduction

The electronic information industry’s demand for cable eccentricity is increasing, driven by the strengthening of the national electronic sector and the evolution of high-performance electronic products. The Chinese national standard GB/T 12706-2020 [1] states that the eccentricity of extruded insulated power cables with rated voltages from 6 KV ($U_{m}=7.2\;KV$) to 30 KV ($U_{m}=36\;KV$) and 35 KV ($U_{m}=40.5\;KV$) must not exceed 15%. The International Electrotechnical Commission’s IEC60502-2 [2] standard similarly dictates that the eccentricity for cables with rated voltages from 6 KV ($U_{m}=7.2\;KV$) to 30 KV ($U_{m}=36\;KV$) must not exceed 15%. These standards highlight the stringent cable eccentricity requirements in both China and the international electronic information industry. The cable-testing industry currently faces challenges with eccentricity-measurement equipment and methods, including low stability, poor accuracy, and high costs. These issues result in the low popularity, poor detection quality, and high costs within the cable manufacturing industry, thus increasing the likelihood of substandard cables reaching the market. The severe eccentricity in these cables can severely impair their mechanical and electrical properties, diminishing the performance and stability of electronic products and potentially hindering the future development of the electronic information industry [3,4,5]. In this context, developing a precise, stable, and cost-effective cable eccentricity-detection method is crucial. This paper proposes a cable eccentricity-detection method based on magnetic field to address the issues of low adoption, accuracy, stability, and high costs in the current cable-inspection industry.

The cable-testing industry predominantly uses X-ray detection, eddy current testing, and optical magnetic detection to obtain cable eccentricity data [5,6,7,8,9,10,11]. X-ray detection within this group offers high stability and accuracy, but it comes with high costs, as advanced X-ray cable eccentricity-testing equipment can cost between USD 10 K and USD 200 K. Operators’ prolonged exposure to X-rays may lead to health issues. The X-ray detection process requires up to 12 steps, making it operationally challenging. X-ray detection equipment has a lower efficiency than the other two methods, testing only 0.1–5 m of cable per second, which may not be suitable for rapid testing scenarios. Eddy current testing is efficient and safe, but it has a limited effective detection range, generally around 1.5 mm, which may not effectively detect eccentricity in large cables. Under strong electromagnetic interference, the error rate of eddy current testing could increase by about 5%, indicating weak anti-interference capabilities. In practical applications, the precision of eddy current testing is 0.1 mm, falling short of the ideal for cable eccentricity detection. Optical magnetic detection of cable eccentricity is precise and efficient, with mature technology, but it is also costly, with equipment priced between USD 10 K and USD 200 K. Therefore, traditional cable eccentricity-detection methods are marred by issues such as poor precision, high costs, low efficiency, high operational barriers, and limited effective detection ranges.

Existing detection equipment, including YN31108-B (EC Plaza: Beijing, China) CENTERVIEW 8000 (SIKORA AG: Bremen, Germany), and ACM-10XY (Adwantek Technologies Co., Ltd: Shenzhen, China), performs high-frequency exposure on cables using optical systems. The equipment projects beam shadows onto charge-coupled device (CCD) sensors and calculates cable eccentricity based on the optical information collected by these sensors. These instruments can achieve high accuracy and efficiency, but they may be influenced by the testing environment, such as changes in light that could lead to deviations in the optical system’s and CCD chip’s alignment, potentially affecting the accuracy of the test results. Additionally, they are characterized by high costs. Chinese devices like XCPY-01 (Guangzhou Rui Ming Instruments Co., Ltd.: Guangzhou, China) and other instruments realize online cable eccentricity detection through X-ray chromatography, high-speed scanning technology, and signal processing. These instruments also achieve precise and efficient detection of cable eccentricity, but they face issues such as potential environmental impact from X-ray radiation, the possible harm to the health of operators, high costs, and complex use processes.

To develop technology that can accurately and efficiently position and range using magnetic field-detection technology in complex working conditions, Chinese and international scholars have conducted in-depth research on cable eccentricity detection using various advanced measurement methods. In this paper, the literature on magnetic induction ranging and positioning technology is thoroughly interpreted and studied. As shown in Table 1, Chen W et al. [12] proposed a nonlinear positioning method for magnetic targets based on the principle of magnetic gradient tensor. They reduced the magnetic interference and array error of the measurement system by combining the ellipsoid fitting method based on the total least squares algorithm. Experiments demonstrated that this method could achieve high-precision positioning and path tracking of magnetic targets. Lei X et al. [13] introduced a linear positioning method based on the two-point magnetic gradient full tensor. They positioned magnetic targets by constructing a two-point magnetic gradient full tensor positioning model and measuring the magnetic gradient full tensor at two positions. Experiments showed that this method reduced detection errors in three magnetic directions and achieved more accurate positioning of magnetic targets. Gaigai L et al. [14] established a two-point magnetic gradient tensor positioning model, derived the dependent equation of the magnetic target position vector, and developed a new magnetic target-positioning method that utilizes only the two-point magnetic gradient tensor without approximation errors. Simulation results indicated that this method had negligible error in the absence of noise and significantly reduced the relative error in noisy environments compared to traditional methods. Experiments further confirmed that the method was unaffected by changes in the distance between two observation points and could mitigate the influence of the geomagnetic field and distance variations. Jihao L et al. [15] presented a novel real-time magnetic dipole positioning method. They located magnetic targets using a cube tensor measurement array based on the scalar triangulation and ranging method. Simulation results revealed that this method significantly expanded the successful positioning area. Li Q et al. [16] designed a second-order magnetic gradient tensor system based on the plane-cross magnetic gradient tensor system. They employed the magnetic gradient tensor space invariant constraint equation, second-order magnetic gradient tensor, and Euler deconvolution method to solve for the magnetic source position coordinates. Both simulation and practical experiments validated that this method could achieve high-precision magnetic source positioning. Xiaoping Y et al. [17] verified the effectiveness and accuracy of injecting low-frequency excitation current into the casing of an adjacent well through time-sharing or frequency-division excitation methods and detecting the magnetic induction intensity generated by the casing current around the formation to determine the distance between wells. Experiments confirmed the effectiveness and accuracy of this method. Binbin D et al. [18] generated magnetic signals by rotating two magnetic particles at fixed positions. They designed two fluxgate sensors to receive the magnetic signals and calculated the distance between the two magnetic particles. Simulation and experimentation proved that this method had high detection accuracy. Weiren C et al. [19] implemented a high-precision geomagnetic orientation technology based on the Mahony algorithm. Experimental results showed that this technology significantly reduced error amplitude, effectively compensated for data distortion, and provided higher and more stable compensation accuracy. Huan L et al. [20] proposed a new two-point tensor positioning method, establishing a two-point magnetic field gradient tensor measurement structure and a target-positioning function with a penalty term for target magnetic moment information. Simulations and experiments have shown that this method can accurately locate magnetic targets stably. Zeng F et al. [21] proposed a magnetic positioning structure for underwater robots, employing a second-order tensor positioning algorithm and a vertical gradient positioning algorithm, combined with the inherent vertical profiling motion of autonomous underwater vehicles, effectively achieving precise positioning of underwater cables. Experiments have demonstrated good underwater positioning effects at close and longitudinal distances with this structure. The aforementioned magnetic ranging and positioning methods demonstrate the feasibility of using magnetic field ranging to measure the distance between the conductor’s center and the magnetic field-collection module, thereby enabling the calculation of cable eccentricity. However, these methods face challenges in accurately detecting cable eccentricity, including issues with meeting industry standards, applicability concerns, and limitations in short-range measurement capabilities. Hence, the application of magnetic ranging technology to the precise detection of cable eccentricity in scenarios, requiring small-distance measurement represents a significant innovation in the cable-inspection industry.

This paper proposes an innovative method for detecting cable eccentricity, based on practical and feasible theories of magnetic positioning and ranging. The method employs magnetic field-detection technology and the principles of electromagnetic induction, utilizing an annular magnetic field array to capture signals. This paper employs a ring array consisting of six magnetic field signal-collection modules, positioned at an optimal distance from the cable surface, to achieve precise short-distance magnetic ranging suitable for cable eccentricity detection. The magnetic field signal-collection modules sample the cable-induced magnetic field signals at high frequencies, facilitating efficient cable eccentricity detection. The cable eccentricity-detection method described samples the ambient magnetic field when the cable is unloaded to establish a no-load value, and then samples again after current is applied to establish a loading value. Subtracting the no-load value from the loading value yields the final value, representing the magnetic flux density B induced by the cable. This approach enhances the precision, stability, and antinterference capabilities of the eccentricity detection. During detection, if a pair of measurement positions yields significantly disparate final values, the system automatically discards this set of data and uses the other two reasonable sets for calculating cable eccentricity, thereby enhancing the accuracy and stability of the detection. The components utilized in this method are all standard types, which are relatively inexpensive, thereby substantially reducing the cost of detection. Furthermore, the higher frequency of magnetic field data acquisition can significantly improve the efficiency of this detection method. The detection method proposed herein, utilizing the aforementioned design, effectively addresses the limitations of existing magnetic ranging technologies and traditional cable eccentricity-detection method for cable eccentricity detection, ensuring its practical feasibility.

## 2. Detection Principle, Finite Element Analysis, and Computational Methods

### 2.1. Principle of Cable Eccentricity Detection

The proposed cable eccentricity-detection method in this paper is capable of conducting eccentricity tests on both single-core and multi-core cables that lack a metallic shield. This method measures the overall eccentricity of the cables. The procedure for inspecting single-core cables is outlined below. When applied to multi-core cables, the method treats the multiple intertwined small cores within the cable as a single large conductor, using the same method as for single-core cables to measure the overall cable eccentricity, which is considered the eccentricity value for the multi-core cable. This technique cannot be applied to inspect cables with metallic shielding layers due to the method’s characteristic of generating a magnetic field inside the cable and detecting it externally. Since the method for detecting multi-core cables by considering the multiple small cores as one large conductor is the same as that for single-core cables, this paper concentrates on presenting the eccentricity-detection method for single-core cables.

As shown in Figure 1a, the copper core conductor in a qualified cable and the cable sheath exhibit a concentric circle relationship. The magnetic field generated by the energized conductor and the sheath forms concentric circles with the conductor’s center as the central point [21,22,23,24]. According to the Biot–Savart Law, an electromagnetic field is generated around an infinitely long straight wire when a steady direct current flows through it. This field is characterized by concentric circles with the conductor’s center as the central point, and the magnetic flux density decreases with increasing radius [25]. The cable detection fixing device and the method for collecting magnetic field information used in this paper ensure that the calculation method for magnetic flux density meets the requirements of an infinitely long straight wire.

As shown Figure 1b, when the cable is in an eccentric condition, the conductor and the protective sheath no longer maintain a concentric circular relationship. Moreover, as indicated in the following ‘Eccentricity Calculation Method’ section, after the cable is energized, the magnetic field acquisition modules at six different positions detect magnetic flux densities that are not identical. Therefore, the eccentricity of the cable can be calculated using the different magnetic flux densities measured at each detection position.

According to Faraday’s law of electromagnetic induction, the formula for calculating the magnetic flux density is as follows [26,27]:(1)$$B=\frac{\mu I}{2\pi d}$$

Using Formula (1), this paper derives the distance formula from the conductor surface to the detection position as follows:(2)$$d=\frac{\mu I}{2\pi B}$$

In the formula, d is the distance from the conductor surface to the detection position. μ is the magnetic permeability. I is the current output by the constant current source to the cable. B represents the magnetic flux density value detected at the detection position of the magnetic field acquisition module of the cable eccentricity-detection instrument described.

Based on Equation (2), the distance d from the set detection positions to the conductor center is obtained. By combining the calculation rules of trigonometric functions, we can deduce the corresponding cable eccentricity data and overall eccentricity information.

As shown in Figure 1c, the cable eccentricity-detection instrument proposed in this paper first secures the cable to be tested onto the detection fixing device. During detection, a constant current source applies an electrical current to the cable, and the magnetic field acquisition module collects the resulting magnetic field signals. Using a ring array, we perform signal collection to minimize acquisition errors and bolster resistance to magnetic interference in signal transmission [28,29,30]. Conditioning circuits then adjust the analog signal voltage from the module to meet the requirements for analog-to-digital conversion, which is subsequently carried out by the converter [31]. The microprocessor processes these digital signals, calculating the magnetic flux density at the acquisition module. Employing computational methods such as the cosine theorem, inverse cosine theorem, and calculus corrections, we determine the distance d from the conductor’s center to the module and calculate the cable conductor’s eccentricity e [9,32,33]. Through serial communication, the human–machine interaction terminal displays the cable eccentricity information [9,34]. Upon receiving parameter adjustment information at the interaction terminal, the detection instrument adjusts its process, facilitating two-way human–machine interaction [35].

### 2.2. Finite Element Simulation and Analysis of Its Results

#### 2.2.1. Finite Element Simulation Experiment

Accroding to the Chinese national standard GB/T 12706-2020 specifies requirements for the conductor radius range of 35 kV medium voltage power cables or below. We select a polyvinyl chloride (PVC/A) cable with a nominal conductor cross-sectional area of 4 mm^2^, a nominal insulation thickness of 1 mm, and a rated voltage of 0.6/1 (1.2) kV as the simulation object. We set the simulation environment with an air domain radius of 20 mm and height of 50 mm, designating a 5 mm distance from the center of the air domain as the lift-off starting point. We simulate the effect of lift-off value on the magnetic flux density of the magnetic field induced by the cable conductor at forward and reverse maximum eccentric positions and in the non-eccentric position during energization, using a loading current of I = 4 A. We also simulate the conductor in the non-eccentric position with currents of I = 2 A, I = 4 A, and I = 6 A to analyze the impact of current on the magnetic flux density of the induced magnetic field.

As shown in Figure 2a, we observe that inside the conductor, the magnetic flux density increases gradually as the distance to the conductor surface decreases. The rate of increase sharpens, reaching its peak at the conductor surface. Outside the conductor, the magnetic flux density decreases gradually with the increasing distance from the conductor surface, and the rate of decrease tapers off.

As shown in Figure 2b,c, we observe that the magnetic flux density at the detection point rapidly decreases initially and then slows down gradually as the lift-off value increases. This behavior conforms to the exponential relationship between distance (d) and magnetic flux density (B), as defined by Equation (2).

As shown in Figure 2b, we observe that changes in the cable conductor’s eccentric position lead to an overall increase or decrease in the magnetic flux density at the detection point. As the conductor moves closer to the detection point, the magnetic flux density increases overall, and the rate of change in the detected magnetic flux density accelerates. Conversely, as the conductor moves farther from the detection point, the magnetic flux density decreases overall, and the rate of change slows down.

As shown in Figure 2c, we see that an increase in current intensity leads to a significant overall rise in magnetic flux density at the detection point. Faster changes occur in areas of high magnetic flux density and slower changes occur at levels below the relative peak of magnetic flux density. As current intensity decreases, the magnetic flux density at the detection point overall decreases, and its rate of change slows down.

#### 2.2.2. Analysis of Finite Element Simulation Experimental Results

Finite element simulation indicates that current intensity and lift-off value are the two primary factors affecting the magnetic flux density values captured by the magnetic field-collection module. These factors influence the magnetic flux density and its trend at the detection location. According to the characteristic of the magnetic sensor, its accuracy in collecting magnetic field information increases with higher ambient magnetic flux density. During detection, increasing the test current and reducing the lift-off value enhance the magnetic flux density at the detection array, which improves the precision of magnetic flux density measurement and the accuracy and reliability of the calculated cable eccentricity (e). Therefore, properly controlling the test current and lift-off value for the cable under test significantly determines the accuracy and reliability of the final detection results. Variations in the distance between the conductor and the magnetic field-collection module directly influence the overall detection values, establishing a theoretical basis for calculating cable eccentricity e with the method outlined in Equation (8). Such evidence confirms the theoretical feasibility of the cable eccentricity-detection method proposed in this paper.

Exploiting the characteristic that energizing a cable alters the magnetic flux density of the environment, we take an initial baseline measurement under unloaded conditions. This initial measurement establishes the no-load value. Post-energization, we measure the magnetic flux density again, which becomes the loaded value. We determine the actual magnetic flux density B by subtracting the no-load value from the loaded value, using this for calculating the cable eccentricity e. We believe that this method can mitigate electromagnetic interference in the detection environment, thereby enhancing the accuracy and stability of our cable eccentricity-detection approach.

### 2.3. Eccentricity Calculation Method

As shown in Figure 3a, we designate the cable sheath surface as the lift-off starting point and set the lift-off value to H = 1 mm. We then simulate the magnetic field by passing a 4 A constant current through the cable in the eccentric state shown in Figure 3b. This process yields three distinct sets of magnetic flux density values, as measured by the magnetic field acquisition modules. Building on these measurements, the cable eccentricity-detection method introduced in this paper calculates the cable’s eccentricity. This calculation integrates computational methods, including the law of electromagnetic induction and the inverse cosine theorem [27,36].

As shown in Figure 3b,c, we employ a cable with an eccentric conductor positioned at the upper right corner as a case study to illustrate the precise calculation method for the cable eccentricity detection. Six magnetic field-collection modules are divided into three groups and arranged in a circular pattern around the cable sheath’s center, maintaining equal spacing circumferentially. We calculate the distance (d) from the conductor’s center to the magnetic field-collection module using Formula (2), incorporating the collected magnetic flux density values. As shown in Figure 3b, ∠1 to ∠6 are the division angles formed by the lines connecting the three vertices of ΔABC to the conductor center point P. As shown in Figure 3b, r signifies the radius of the conductor. The coordinates (x,y) in Figure 3b pinpoint the conductor’s center point p within the depicted coordinate system. h denotes the horizontal position of the magnetic field acquisition module at the detection point within the shown coordinate system. R stands for the radius of the cable sheath. H is the lift-off value specified in this work. $t_{max}$ (GE) and $t_{min}$ (FH), as shown in Figure 3b, represent the cable sheath’s maximum and minimum thicknesses, respectively. The formula commonly utilized in the cable-testing industry for eccentricity calculations is [4]:(3)$$e=\frac{t_{max}-t_{min}}{t_{max}}$$

In the formula, e represents the cable eccentricity, $t_{max}$ denotes the maximum thickness of the cable sheath, and $t_{min}$ indicates the minimum thickness of the cable sheath.

As shown in Figure 3d,e, we illustrate the cable eccentricity-detection method detailed in this paper, which calculates the distance d from the module to the conductor center, following the detection of magnetic flux density at the magnetic field acquisition module’s location. We then apply the angular relationships presented in Figure 3b, with the conditions ∠1 + ∠2 = 30°, ∠3 + ∠4 = 60°, and ∠5 + ∠6 = 90°, using the Law of Cosines and the cosine sum formula [37,38].
(4)$$\cos A=\frac{b^{2}+c^{2}-a^{2}}{2bc}$$
(5)$$\cos\left(A+B\right)=\cos A\times \cos B-\sin A\times \sin B$$

In the formula, A denotes the measure of an arbitrary angle within any triangle. B denotes the measure of a arbitrary angle within any triangle. a, b and c represent the sides of the same triangle that are opposite to angles ∠A, ∠B, and ∠C, respectively.

We apply Equations (4) and (5), along with the cosine rule, to establish the following relationships for the angles: cos⁡30°=cos⁡(∠1+∠2), cos⁡60°=cos⁡(∠3+∠4), cos⁡90°=cos⁡(∠5+∠6). Using the computational methods of the cosine rule, we deduce the coordinates (x, y) of the conductor center point P [39]. The x and y values represent the displacements of the conductor center in the X and Y directions, ΔX and ΔY, respectively. We then apply the Pythagorean theorem [37]:(6)$$a^{2}+b^{2}=c^{2}$$

In the formula, a is one leg of a right-angled triangle, b is the other leg of the same right-angled triangle, and c is the hypotenuse of the same right-angled triangle.

We calculate the eccentric distance ∆E using Equation (6) and then compare it with the conductor radius r to determine the maximum and minimum sheath thickness of the cable. The comparison formula is:(7)$$\left\{\begin{array}{c} t_{max}=R+\left(∆E-r\right),\;t_{min}=R-\left(∆E+r\right),∆E>r\\ t_{max}=R+\left(r-∆E\right),\;t_{min}=R-\left(∆E+r\right),r>∆E \end{array}\right.$$

In the formula, $t_{max}$ represents the maximum thickness of the cable sheath, $t_{min}$ represents the minimum thickness of the cable sheath, +R denotes the radius of the cable sheath, r denotes the radius of the conductor, and ∆E denotes the eccentric distance of the conductor.

Using the computational formulas mentioned earlier, we derive a general system of equations that connects the magnetic flux density B, as detected by the magnetic field-collection modules, to the eccentricity e. In practice, our magnetic field-based cable eccentricity-detection method utilizes this system of equations, specifically Equation (8), to calculate e from the B values measured by the six modules:(8)$$\left\{\begin{array}{c} d=\frac{\mu I}{2\pi B}\\ \cos∠1=\frac{\left(x^{2}+y^{2}\right)+{\left(\frac{2h}{\sqrt{3}}\right)}^{2}-d^{2}}{\frac{4h}{\sqrt{3}}\times \sqrt{x^{2}+y^{2}}}\\ \cos∠2=\frac{x}{\sqrt{x^{2}+y^{2}}}\\ \cos∠3=\frac{{\left(\frac{2h}{\sqrt{3}}\right)}^{2}+d^{2}-\left(x^{2}+y^{2}\right)}{\frac{4h}{\sqrt{3}}\times d}\\ \cos∠4=\frac{d^{2}+\frac{h^{2}}{3}-\left({\left(h-x\right)}^{2}+y^{2}\right)}{\frac{2h}{\sqrt{3}}\times d}\\ \cos\left(∠1+∠2\right)=\cos∠A\\ \cos\left(∠3+∠4\right)=\cos∠B\\ t_{max}=R+\left(∆E-r\right),t_{min}=R-\left(∆E+r\right),∆E>r\\ t_{max}=R+\left(r-∆E\right),t_{min}=R-\left(∆E+r\right),r>∆E\\ e=\frac{t_{max}-t_{min}}{t_{max}} \end{array}\right.$$

In the formula, d represents the distance from the conductor to the detection point. μ is the magnetic permeability of the medium. I is the current intensity applied to the cable. B is the magnetic flux density value detected by the magnetic field-collection module. h is the horizontal coordinate in the coordinate system of the detection position where the magnetic field-collection module is located. x and y are the horizontal coordinates of the conductor center point P in the coordinate system. ∠1 to 4 are the calculated division angles in Figure 3b for ∠A and ∠B in triangle ΔABC, divided by the lines from points A and B to the conductor center point P. $t_{max}$ is the maximum thickness of the cable sheath. $t_{min}$ is the minimum thickness of the cable sheath. R is the radius of the cable sheath. r is the conductor radius. ∆E is the eccentric distance of the conductor. and e is the cable eccentricity.

Based on Equation (8), we secure the cable under test in the fixture shown in Figure 1c to ascertain the h value at the detection position. After obtaining the magnetic flux density value B, we calculate the distance d from the detection position to the conductor. Based on the Law of Cosines, we calculate the cosine values of any two sets of divided angles (for example, if choosing ∠B and ∠C, the divided angles are ∠3 to ∠6). In Equation (8), we select and calculate the cosine values of cos⁡∠1 to cos⁡∠4, which directly correspond to the cosine values of the two sets of divided angles for ∠A and ∠B, as shown in Figure 3b. Using the Pythagorean theorem, we derive the corresponding sine values from the calculated cosine values. Based on the known values of cos⁡∠A=cos⁡30° and cos⁡∠B=cos⁡60°, we apply the cosine sum theorem to substitute the cosine and sine values of the two sets of divided angles. Utilizing the MCU’s numerical computation library, we employ numerical methods to solve the simultaneous equations, specifically cos⁡∠A=cos⁡30°=cos⁡(∠1+∠2) and cos⁡∠B=cos⁡60°=cos⁡(∠3+∠4). Through this process, we determine the x and y coordinates of the conductor’s center point P. Using the Pythagorean theorem and the values of x and y, we calculate the conductor’s eccentricity distance ∆E. Considering the relationship between the eccentricity distance ∆E and the conductor radius r, we choose appropriate calculation methods to determine $t_{max}$ and $t_{min}$, which correspond to GE and FH in Figure 3b. Employing the values of $t_{max}$ and $t_{min}$, we determine the final cable eccentricity e using the formula outlined in Equation (8).

Furthermore, the calculation process mentioned above applies to each detection position where the six magnetic field acquisition modules are situated. At each position, the module can independently calculate an eccentricity value e using the aforementioned process. By comparing these calculated values, we eliminate outliers with errors or excessive deviations. This refinement allows us to determine the final cable eccentricity values and draw accurate data conclusions.

## 3. Experimental Results and Analysis

According to the national standard GB/T 12706-2020.1. Power cable with extruded insulation with rated voltage of 1 kV ($U_{m}=1.2\;kV$) to 35 kV ($U_{m}=40.5\;kV$) and accessories—Part 1: Cable with rated voltage of 1 kV ($U_{m}=1.2\;kV$) to 3 kV ($U_{m}=3.6\;kV$), a polyvinyl chloride (PVC/A) cable with nominal conductor cross-sectional area of $4{mm}^{2}$, nominal thickness of cable insulating sheath of 1 mm and rated voltage of 0.6/1 (1.2 kV) was selected as the cable to be detected. The eccentricity-detection experiment is conducted with the surface of the cable sheath as the starting point for lift-off. After the detection experiment is concluded, the cable sheath of the subject under test is cut open to measure the actual cable eccentricity information for comparative analysis. After completing the calibration experiment, lift-off value influence experiment, and current intensity influence experiment, the cable sheath was cut open and the actual eccentricity e was measured to be 7.5%.

### 3.1. Calibration Experiment

To verify the accuracy of the aforementioned eccentricity-detection method, a calibration experiment is designed. As shown in Figure 4, the experiment employs a height gauge to establish lift-off values of H = 1 mm and H = 2 mm for the calibration test points. During the calibration experiment, a current of I = 4 A is selected for application to the cable. The experiment consists of two sets, with each set repeated ten times. The experimental procedure is shown in Figure 4. The experiment produces a data variation chart showcasing the results of repeated experiments under specified conditions at lift-off values of H = 1 mm and H = 2 mm, as shown in Figure 5 and Table 2, which compares the experimental data and errors for experiments conducted with the two distinct lift-off values.

As show in Figure 5a and Table 2, the detected eccentricity at the detection position with a lift-off value H = 1 mm, using the actual measured eccentricity as the benchmark, varies between 0 and 0.3% above and below the standard line across ten experiments. The error ranges from 0 to 4%, and the numerical variation standard deviation is 0.11. Figure 5b illustrates that in the ten experiments conducted at the detection position with a lift-off value H = 2 mm, the detected cable eccentricity varies between 0.2% and 0.5% above and below the standard line. The error ranges from 2.7% to 6.7%, and the numerical variation standard deviation is 0.09.

The experimental results clearly show that, under the specified conditions of lift-off value and applied loading current, the detected cable eccentricity exhibits a small error when compared to the actual standard eccentricity. The variation curve tightly adheres to the standard value, and the standard deviation of numerical fluctuations across the 10 repeated detection experiments remains small, indicating no significant deviations. This underscores the method’s high detection accuracy for cable eccentricity and its ability to maintain robust detection stability across multiple repetitions, thereby guaranteeing the reliability of the detection data.

### 3.2. Lift-Off Value Influence Experiment

To investigate the impact of lift-off value on the detection accuracy of the cable eccentricity-detection method introduced in this paper, a lift height influence experiment is designed. The experiment sets the loading current intensity at I = 4 A and employs a height gauge to perform experiments across the lift height range of H = 0 to 4 mm, with an experimental increment of 0.2 mm, as shown in Figure 4. Conducting the lift-off value influence experiment results in the acquisition of the error bar chart presented in Figure 6, which visually represents the variation of detection error with respect to the lift-off value. Additionally, Table 3 provides the eccentricity-detection data and corresponding errors under various lift-off value conditions.

As shown in Figure 6 and Table 3, the different lift-off positions in this experiment significantly affect the accuracy of cable eccentricity detection. When the lift-off value is small, the detection accuracy for cable eccentricity is relatively high, whereas a larger lift-off value results in lower detection accuracy. The detection accuracy of cable eccentricity decreases with the increase in lift-off value, exhibiting an overall linear upward trend within the detection range, which aligns with the trend demonstrated by Equation (2). Within the lift-off value range set in this experiment, the detected eccentricity values are within 0.03~0.76% of the standard detection values, with an error range of 0~9%.

The cause of detection error is primarily attributed to the increase in lift-off value, which leads to a reduction in the magnetic flux density values at the detection position excited by the electrified cable, which results in decreased accuracy of the magnetic field data detection, which in turn causes a decrease in the precision of the determined cable eccentricity. Therefore, the detection error of cable eccentricity increases as the lift-off value becomes larger.

The results of the lift-off value influence experiment demonstrate that Cable eccentricity-detection method based on magnetic field proposed in this paper is subject to varying degrees of change due to the variations in the magnetic flux density values at the detection position of the magnetic field acquisition module, which are induced by changes in the lift-off value. The detection accuracy decreases as the lift-off value increases. The detection error is maintained within a low range, confirming that the described cable eccentricity-detection method possesses high detection precision. This method is capable of achieving high-precision detection across a broad range of lift-off values, does not demand stringent detection distance requirements, and has a wide scope of practical application with a low entry barrier for use.

### 3.3. Current Intensity Influence Experiment

To investigate the impact of the current intensity applied to the cable on the accuracy and stability of the cable eccentricity detection, a current intensity influence experiment was designed. The loading currents were set to I = 2 A, 3 A, 4 A, 5 A, and 6 A, with a lift-off value H = 1 mm. As depicted in Figure 7, in this experiment, the cable was clamped onto the described detection fixture, and after applying different intensities of current, the designed eccentricity-detection instrument was used to perform the eccentricity detection. Using the human–machine interaction terminal to display the detection information, and receive the detection status adjustment information inputted. The experiment yielded the cable eccentricity variation chart under different currents, as shown in Figure 7 and the Table 4 showing the eccentricity-detection data and error under different current intensity conditions.

As shown in Figure 7 and Table 4, with the increase in the current intensity applied to the cable, the detection error for cable eccentricity decreases and the detection accuracy improves, exhibiting a linear trend of change within the detection range. Based on the actual measured values as the standard, the detected eccentricity values vary between 0.07~0.53% of the standard values, with errors ranging from 0.9 to 7%.

The primary source of error in this experiment is mainly due to the variation in the current intensity applied to the cable, which causes changes in the magnetic flux density values at the location of the magnetic field acquisition module. This leads to varying degrees of deviation in the collected magnetic field data, ultimately resulting in errors in the detected cable eccentricity measurements.

Our experiments show that cable eccentricity-detection accuracy changes with the variation in the current intensity applied to the cable. Increasing the current intensity excites a greater magnetic flux density, enhancing the magnetic field’s detectability at the magnetic field acquisition module’s position. This facilitates easier collection of magnetic field information, leading to improved detection accuracy. The magnetic field-based cable eccentricity-detection method we propose offers high accuracy across various current conditions. It maintains low error levels under different current intensities, demonstrating its capability to accurately detect cable eccentricity with a broad range of current conditions. The method’s relaxed detection conditions enhance its practicality and can potentially reduce the costs associated with cable eccentricity detection.

Above all, the cable eccentricity-detection method we propose is influenced by both the current intensity and the lift-off value. These factors jointly determine the magnetic flux density at the detection point of the magnetic field acquisition module, which in turn affects the detection accuracy of cable eccentricity. Our method demonstrates high detection accuracy and stability. It is adaptable to a variety of detection conditions, offering practical advantages and the potential to lower detection costs.

## 4. Error Analysis

### 4.1. Current Intensity and Lift-Off Value Influence

Experiments show that the magnetic flux density at the detection site, essential for accurate results, is significantly influenced by the current intensity and lift-off value. These two parameters are the main factors affecting our cable eccentricity-detection method. By increasing the current intensity and reducing the lift-off value, we enhance the magnetic flux density around the magnetic field-collection array. This leads to more precise and reliable measurements of cable eccentricity, ensuring better stability across a series of repetitive tests. Improper settings for current and lift-off value can cause significant errors, diminishing the reliability of the results. In practice, the current applied to the cable is usually set between half and the full rated current, and the lift-off value, measured from the cable surface, is generally set between 1–4 mm for optimal results. In actual detection, we first determine the lift-off value and then adjust the applied current based on the detection effects to achieve the desired outcome. For example, in the case of a polyvinyl chloride (PVC/A) cable with a rated voltage of 0.6/1 (1.2) kV, a typical rated current of 100 A, and a conductor cross-sectional area of 25 square millimeters, a lift-off value of 1–5 mm is generally chosen, and a current ranging from 50 A to 100 A is set as the test loading current for the detection of cable eccentricity.

### 4.2. Sensor Error Impact

In actual detection scenarios, we expect that one of the six sensors in the magnetic field-collection array might develop errors, a common occurrence. During practical detection, if a sensor shows a significant error, it will be evident as the distance-measurement data at the sensor’s position and its opposing position in the array deviating from the cable’s specified parameters For example, if the sum of the conductor’s distance to the left sheath closing to the left detection position (as shown at position “2’“ in Figure 3b) and the right sheath closing to the right detection position (as shown at position “2” in Figure 3b) greatly exceeds the conductor’s diameter plus the cable’s specified cross-sectional dimensions, our cable eccentricity-detection system will automatically reject this set of data. It will then assess the conformity of the data output from the remaining two pairs of positions with the standard cross-sectional parameters of the cable and select the data set with the best fit as the basis for the final eccentricity calculation. This approach ensures the accuracy and reliability of the eccentricity detection by removing outliers that could distort the results.

### 4.3. Environmental Magnetic Field Impact

In actual detection, we must address magnetic field disturbances from the Earth’s magnetic field, permanent magnets in other moving motors, and surrounding cables carrying high currents, as they can cause significant electromagnetic interference during the cable eccentricity-detection process. This interference can affect the accuracy of our measurements. We must account for these disturbances when conducting cable eccentricity tests to ensure reliable and precise results. Our cable eccentricity-detection method starts with sampling the magnetic flux density in the environment when the cable is not energized, using this as the no-load value. After energizing the cable, we conduct another sampling to obtain the loaded value. We determine the actual magnetic flux density B induced by the energized cable by subtracting the no-load value from the loaded value, which is then used to calculate the cable’s eccentricity. This process is crucial for accurately determining the degree of conductor’s deviation from the cable’s center. Our method can eliminate or reduce the impact of environmental interference, enhancing the detection accuracy and environmental adaptability of the cable eccentricity-detection process.

## 5. Conclusions

This paper introduces a magnetic field-based cable eccentricity-detection method. Utilizing electromagnetic induction principles, we employ a ring-shaped detection array made up of 3 pairs of magnetic field acquisition modules to measure the magnetic field generated by the electrified cable. We calculate eccentricity using the cosine rules in combination with angular relationships. Our simulations and experiments show that this method is affected by the lift-off value and current intensity. In practical applications, controlling these two factors is essential for high-quality detection results. The method we propose delivers high-precision detection effects, with high stability and reliability of results. It also demonstrates strong adaptability to various detection conditions and is both theoretically and practically feasible for cable eccentricity detection. Additionally, it can reduce detection costs. This paper innovatively presents a magnetic field-based cable eccentricity-detection method, offering a new approach to cable inspection for the cable quality control industry and contributing to the advancement of the electronic information industry.

This paper introduces a magnetic-based cable eccentricity-detection method applicable not only in cable testing but also across various fields. For example, in robotics, we can use array-based magnetic ranging technology for robot navigation and positioning. In engineering, the technology helps measure the distances between adjacent wells in drilling projects. In the medical device sector, it assists in determining the location of wireless capsule endoscopes and monitoring their orientation within the body. The cable conductor position analytical algorithm, which employs the law of cosines and numerical method, discussed in this paper, can also be applied to robotic motion posture analysis. In conclusion, the techniques of our detection method offer diverse applications across multiple industries, enhancing their commercial value with high versatility, practicality, and commercial potential.

## Figures and Tables

**Figure 1 sensors-24-05525-f001:**
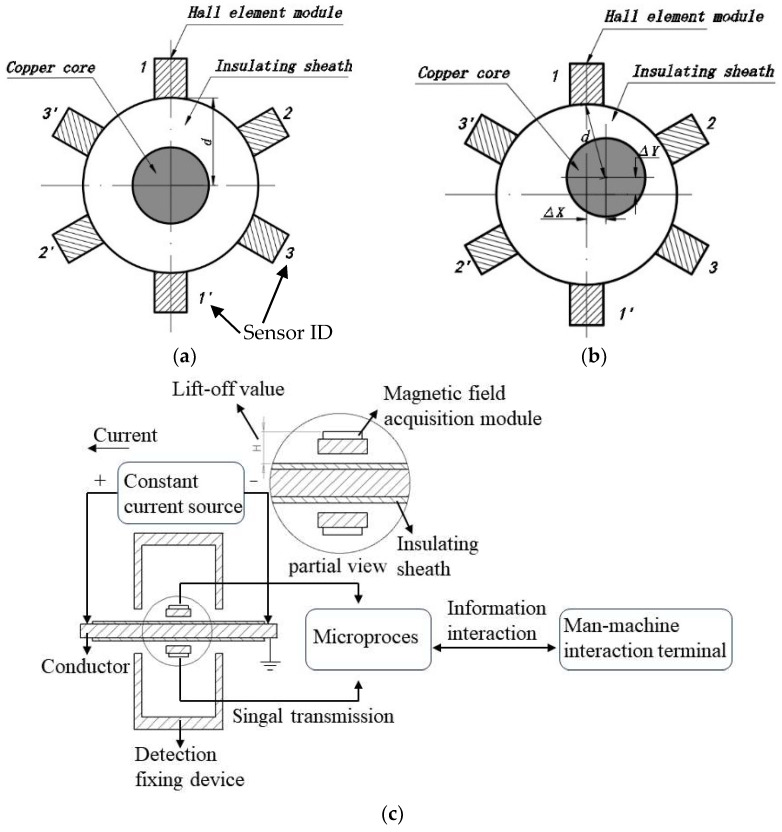
Eccentricity-detection principle diagram. (**a**) Schematic of the inspection cross-section for qualified cables. (**b**) Schematic of the inspection cross-section for eccentric cables. (**c**) Working principle diagram of the cable eccentricity-detection instrument.

**Figure 2 sensors-24-05525-f002:**
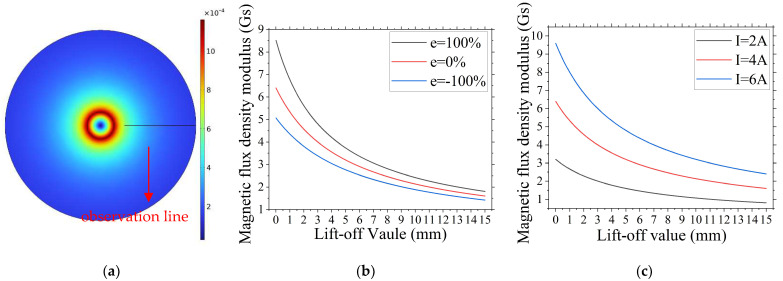
Finite element analysis diagram. (**a**) Magnetic flux density cross-sectional simulation chart. (**b**) Magnetic flux density variation chart under different eccentric conditions. (**c**) Magnetic flux density variation chart under different current intensities.

**Figure 3 sensors-24-05525-f003:**
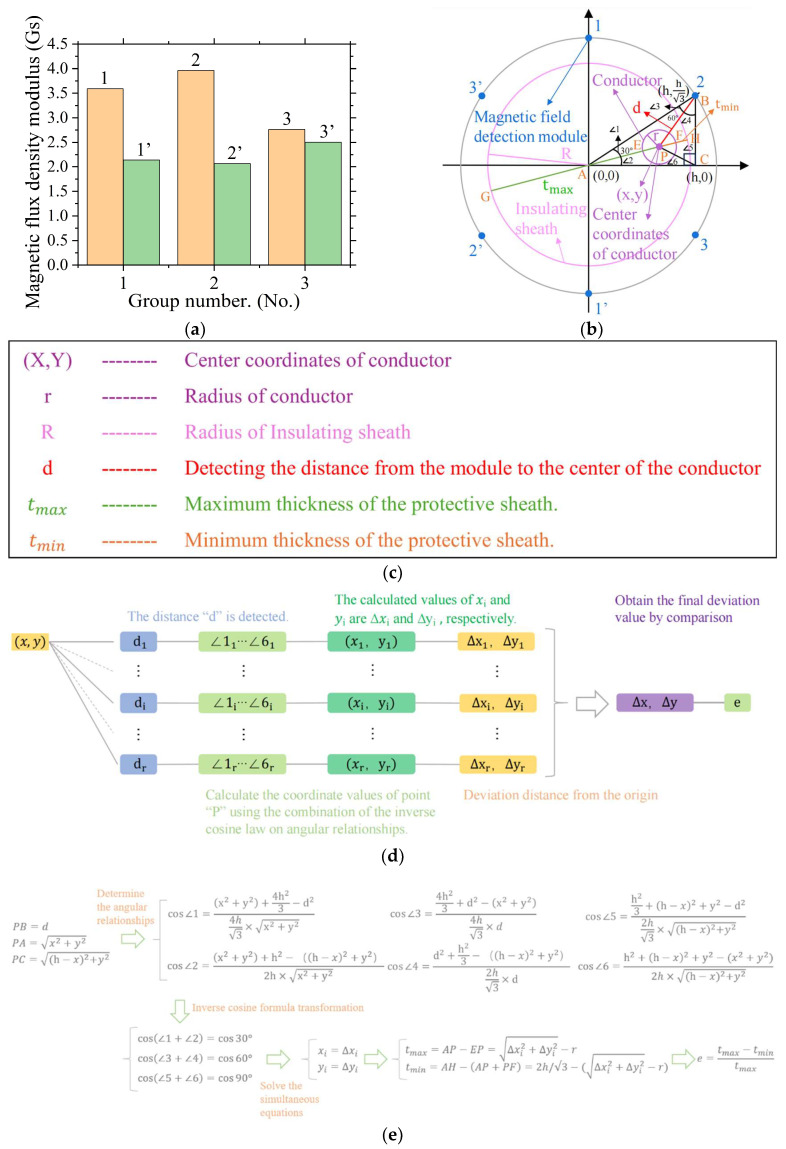
Schematic diagram of the calculation method. (**a**) Detection data graph at a lift-off value t = 1 mm. (**b**) Coordinate system schematic for eccentricity calculation. (**c**) Legend of symbols in the calculation schematic. (**d**) Flowchart for overall eccentricity calculation. (**e**) Flowchart for detailed eccentricity calculation.

**Figure 4 sensors-24-05525-f004:**
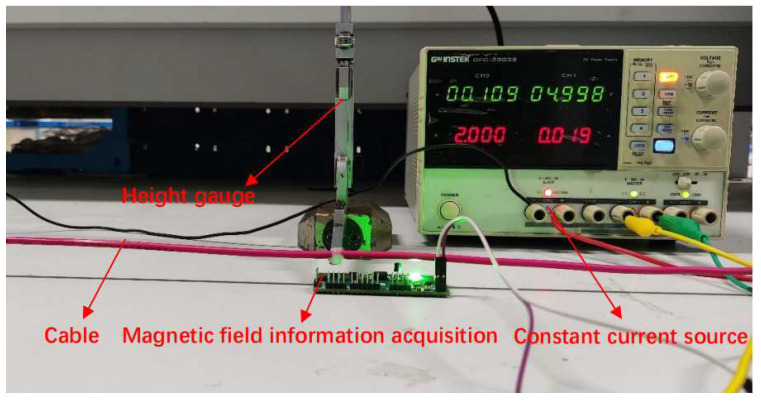
Calibration experiment procedure diagram.

**Figure 5 sensors-24-05525-f005:**
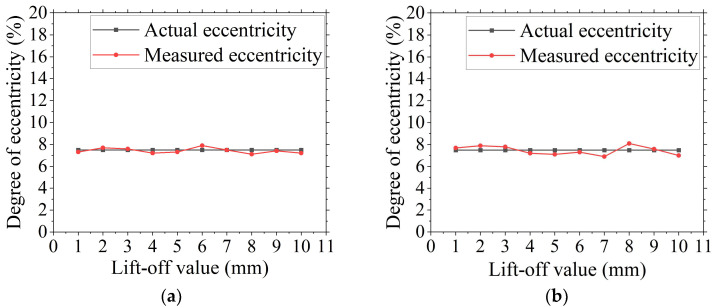
Calibration experiment data chart. (**a**) Detection data variation chart at lift-off height H=1 mm. (**b**) Detection data variation chart at lift-off height H=2 mm.

**Figure 6 sensors-24-05525-f006:**
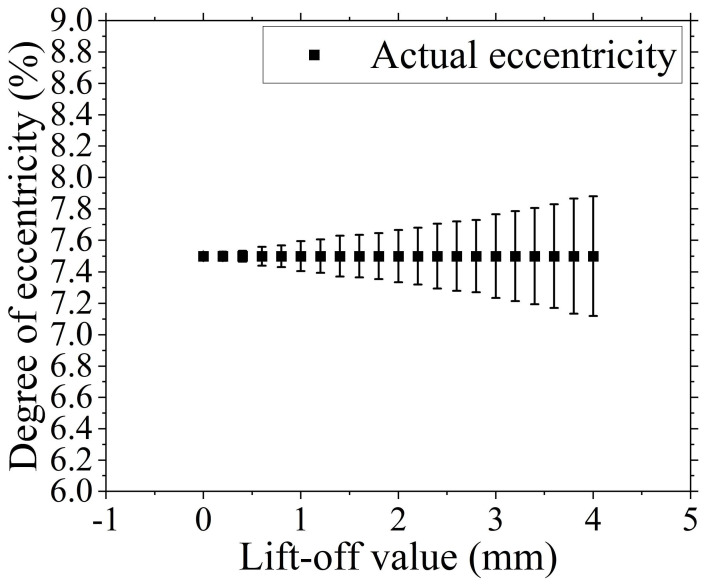
Displaying the manner and trend of change in detection accuracy as the lift-off value increases.

**Figure 7 sensors-24-05525-f007:**
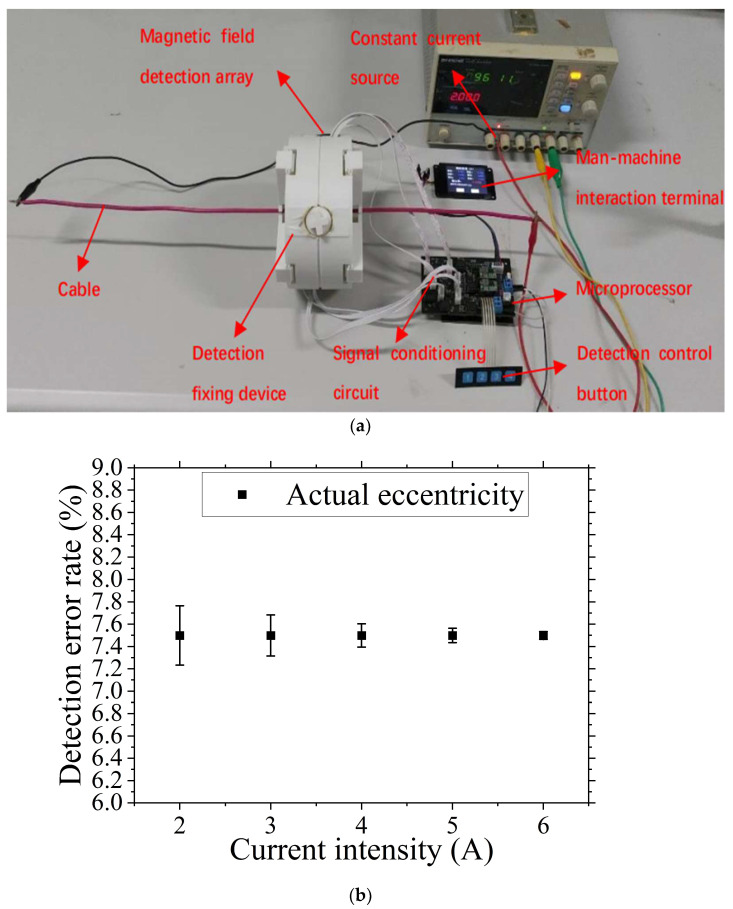
Current intensity influence experiment process and results. (**a**) Current intensity influence experiment process diagram. (**b**) Detection accuracy variation chart under different current intensities.

**Table 1 sensors-24-05525-t001:** Comparison of existing magnetic ranging and positioning methods.

Researcher	Topic	Principle	Conclusions	Deficiencies
Chen W et al. [12]	Magnetic target nonlinear positioning method	An ellipsoid fitting method based on the total least square algorithm	The technology can achieve high-precision path tracking and positioning of magnetic targets	The detection accuracy of this method does not meet the small-distance measurement requirements for cable eccentricity detection
Lei X et al. [13]	Magnetic target linear location Method	A linear location method based on the two-point magnetic gradient full tensor	The method reduces the relative error percentages in the three magnet field directions	The error in the Y-direction is relatively large and there is insufficient experimental support
Liu GG et al. [14]	Magnetic target localization method	A new magnetic target localization method using only a two-point magnetic gradient tensor and no approximation errors	The method can avoid the influence of the geomagnetic field and the variation in the distance, and achieve high localization accuracy	The detection error of this method is too high for the eccentricity detection of cables
Jihao L et al. [15]	Magnetic dipole localization method	The cube tensor measurement array from the scalar triangulation and ranging method	The method will expand the area of successful localization	There is insufficient experimental support
Li Q et al. [16]	Magnetic object positioning	The constraint equation of magnetic gradient tensor spatial invariants, the second-order magnetic gradient tensor and Euler deconvolution	The method realizes a small magnet pinpoint positioning in a 2.1 m heading-line detection interval	The detection accuracy of this method does not meet the small-distance measurement requirements for cable eccentricity detection
Yan Xiaoping et al. [17]	Active magnetic ranging method	A new method of ranging between adjacent wells based on the casing current excitation of adjacent wells	The method solves the problem of ranging of multiple adjacent wells in cluster well development, thus reducing the risks and costs of oilfield development without affecting the operation of adjacent wells	The method is mainly intended for distance measurement of large structures and is not applicable for the precise short-distance measurement required in cable detection
Binbin D et al. [18]	An algorithm to improve magnetic ranging accuracy for cluster horizontal wells with narrow spacings	A new ranging algorithm for the two sensor packages-rotating magnet ranging system	Applying this algorithm in the field can successfully aid in controlling the spacing of cluster horizontal wells more accurately	The method is mainly intended for distance measurement of large structures and is not applicable for the precise short-distance measurement required in cable detection
Chen WR et al. [19]	Magnetic field interference correction of high-precision geomagnetic directional technology	A combined correction model and a high-precision geomagnetic directional technology	The method reduces the error amplitude and effectively compensates the data distortion and can be applicated in military, industrial and civilian fields	The method is primarily used for angle detection and is not suitable for precise short-distance measurement in cable detection
Huan L et al. [20]	Magnetic dipole two-point tensor positioning	A new two-point tensor positioning method based on a magnetic moment constraint	This method enables the localization and tracking of magnetic targets even when sensor precision is relatively low or the environmental noise is high	The method’s detection approach is not suitable for the eccentricity detection of cables
Zeng F et al. [21]	Magnetic gradient tensor positioning method	Second-order tensor positioning algorithms and vertical gradient positioning algorithms	The accuracy of vertical magnetic field detection is higher than that of horizontal magnetic field detection	The detection accuracy of this method does not meet the small-distance measurement requirements for cable eccentricity detection

**Table 2 sensors-24-05525-t002:** Comparison of experimental data and error for experiments with 2 different lift-off values.

Inspection Number	Eccentricity at H = 1 mm	Error Rate	Eccentricity at H = 2 mm	Error Rate
1	7.3%	2.7%	7.7%	2.7%
2	7.7%	2.7%	7.2%	4%
3	7.6%	1.3%	7.9%	5.3%
4	7.2%	4%	7.2%	4%
5	7.3%	2.7%	7.1%	5.3%
6	7.5%	0%	7.2%	4%
7	7.5%	0%	7.2%	4%
8	7.2%	4%	7.8%	4%
9	7.4%	1.3%	7.7%	2.7%
0	7.2%	4%	7%	6.7%

**Table 3 sensors-24-05525-t003:** Lift-off value influence experiment: eccentricity-detection data and error table.

Lift-Off Value (mm)	Rate of Eccentricity	Error Rate
0	7.53%	0.4%
0.2	7.56%	0.8%
0.4	7.57%	0.9%
0.6	7.62%	1.6%
0.8	7.64%	1.8%
1	7.69%	2.5%
1.2	7.71%	2.8%
1.4	7.76%	3.5%
1.6	7.77%	3.6%
1.8	7.79%	3.8%
2	7.83%	4.4%
2.2	7.86%	4.8%
2.4	7.91%	5.4%
2.6	7.9%	5.8%
2.8	7.96%	6.1%
3	8.03%	7%
3.2	8.07%	7.6%
3.4	8.11%	8.1%
3.6	8.16%	8.8%
3.8	8.23%	9.7%
4	8.26%	10.1%

**Table 4 sensors-24-05525-t004:** Current intensity influence experiment: eccentricity-detection data and error table.

Current Intensity (A)	Rate of Eccentricity	Error Rate
2	8.0%	7.1%
3	7.9%	4.9%
4	7.7%	2.8%
5	7.6%	1.7%
6	7.6%	0.9%

## Data Availability

The data presented in this study are available on request from the first author.

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
