# Peer review of "Cable Eccentricity Detection Method Based on Magnetic Field"

_sensors, 2024, doi:10.3390/s24175525_

Round 1

Reviewer 1 Report

Comments and Suggestions for Authors
  • 1. The abstract concisely summarizes the main contributions of the paper. However, it could be slightly improved by highlighting the key advantages of the proposed method more explicitly, such as the reduction in detection costs, higher precision, and improved stability.
  • 2.It would be beneficial to mention the limitations of existing methods in more detail, specifically pointing out how the proposed method addresses those limitations.
  • 3. Consider adding a discussion of the assumptions made in the finite element model and their potential impact on the results.
  • 4.A more in-depth discussion of the sources of error and potential avenues for improving the accuracy of the method would strengthen the paper.
  • 5. The conclusions summarize the main findings effectively. However, it would be valuable to explicitly state the practical implications of the research, such as potential applications in industry and the potential for commercialization.
Comments on the Quality of English Language

The quality of the English language in the document "sensors-3128793" is generally good, with a clear and concise writing style. However, there are a few minor points that could be improved to further enhance the readability and clarity of the text:

1. Some phrases could be simplified to avoid redundancy. For example, "the magnetic flux density cross-sectional simulation diagram as depicted in Figure 2(a)" could be rephrased as "the magnetic flux density cross-sectional simulation diagram in Figure 2(a)"。

2. Most of the document uses the present tense consistently, which is appropriate for describing a method. However, occasionally the past tense is used in describing the method (e.g., "As depicted in the detection depicted in the detection cross-sectional diagram..."). Ensuring consistency in tense throughout the document would make it more cohesive.

Author Response

August 9, 2024

Dear Editors and Reviewer,

  Thank you for your letter and for the reviewers’ comments concerning our manuscript entitled “Cable Eccentricity Detection Method Based on Magnetic Field” (ID: sensors-3128793). Those comments are all valuable and very helpful for revising and improving our paper, as well as the important guiding significance to our researches. We have studied comments carefully and have made correction which we hope meet with approval. Revised portion are marked with color of yellow in the paper. The main corrections in the paper and the responds to the reviewer’s comments are as flowing:

Comments 1:The abstract concisely summarizes the main contributions of the paper. However, it could be slightly improved by highlighting the key advantages of the proposed method more explicitly, such as the reduction in detection costs, higher precision, and improved stability.

Response 1:Thank you for your valuable insights. We agree with this comment. We have explicitly highlighted in the abstract the advantages of our method over traditional methods, such as reduced detection costs, enhanced precision, and improved reliability and stability.The specific location in the ppaper is line 14-16, page 1, highlighted in yellow for emphasis.

Comments 2:It would be beneficial to mention the limitations of existing methods in more detail, specifically pointing out how the proposed method addresses those limitations.

Response 2:Thank you for your valuable insights. We agree with this comment. In this version of the manuscript, we have listed specific data such as the high equipment cost of X-ray detection, ranging from $10K to $200K, and the detection speed of 0.5-5 m/s, providing a detailed explanation of the deficiencies in traditional X-ray detection, eddy current detection, and optical magnetic detection, including high costs, low efficiency, poor accuracy, and poor stability. Additionally, we have elucidated the limitations of existing magnetic ranging and positioning technologies in applicability to the precise detection of cable eccentricity in small-distance scenarios. Specifically, we have outlined the solutions offered by our method in terms of stability, detection accuracy, cost, and efficiency, addressing the limitations present in current magnetic ranging methods and traditional cable eccentricity detection methods.The specific location in the paper is line 45-61, page 2、line 127-160, page 3-5, highlighted in yellow for emphasis.

Comments 3:Consider adding a discussion of the assumptions made in the finite element model and their potential impact on the results.

Response 3:Thank you for your valuable insights. We agree with this comment. In terms of the assumptions made for finite element simulation, we have added the following assumption items:

1) An assumption regarding the impact of current intensity and lift-off value on the accuracy of cable eccentricity detection and the underlying causes.

2) An assumption that based on the phenomenon of the magnetic flux density values at the six detection positions changing with the conductor's position, our method can calculate the cable eccentricity by combining this characteristic with the law of electromagnetic induction.

3) By utilizing the characteristic that the magnetic flux density of the surrounding magnetic field changes when the cable is energized, a magnetic flux density measurement is conducted on the environment in an unloaded state before the test cable is energized. The measured value at this time is taken as the no-load value, and the magnetic flux density value detected after loading is taken as the loaded value. Subtracting the no-load value from the loaded value, the resulting data is used as the actual magnetic field's magnetic flux density modulus excited by the cable's current for subsequent cable eccentricity calculations. We make a assumption that This method can improve the accuracy and stability of detection and better adapt to the detection environment.

The specific location in the paper is line 258-281, page 8, highlighted in yellow for emphasis.

Comments 4:A more in-depth discussion of the sources of error and potential avenues for improving the accuracy of the method would strengthen the paper.

Response 4:Thank you for your valuable insights. We agree with this comment. In terms of error analysis, we have identified the current intensity, lift-off value, environmental electromagnetic interference, and sensor errors as the main sources of error. For these error sources, the following analyses have been conducted in the paper:

1) The error sources of current intensity and lift-off value are primarily due to the initial presets of the current and lift-off value being unreasonable, which leads to a decrease in the accuracy of eccentricity detection. The solution is to determine the current intensity between half of the cable's rated current and the rated current after setting the lift-off value within 1-5 mm to achieve the expected detection results.

2) The environmental electromagnetic interference primarily originates from the Earth's magnetic field, the magnets of nearby motors, etc. The solution is to conduct a magnetic flux density measurement of the environment in an unloaded state before the test cable is energized, using the measured value as the no-load value, and the magnetic flux density value detected after loading as the loaded value. The data obtained by subtracting the no-load value from the loaded value is used as the actual magnetic flux density modulus excited by the cable's current for subsequent cable eccentricity calculations.

3) Sensor errors are a common occurrence, which can be caused by the placement of the detection device, the applied pressure, temperature, and other factors. The solution is to configure the cable eccentricity detection system such that after calculating the distance from the two conductors at the same pair of detection positions to the protective sheath boundary, the data set is compared with the standard cross-sectional dimensions of the cable to be tested. If there is a significant discrepancy, this data set can be discarded, and the remaining qualified data can be used for cable eccentricity calculation.

The specific location in the paper is line 511-558, page 17-18, highlighted in yellow for emphasis.

Comments 5:The conclusions summarize the main findings effectively. However, it would be valuable to explicitly state the practical implications of the research, such as potential applications in industry and the potential for commercialization.

Response 5:Thank you for your valuable insights. We agree with this comment. In terms of multi-domain application value, the paper illustrates that the core technologies of the described cable eccentricity detection method—precision ranging technology for short distances and spatial positioning algorithms—have applications across various fields. The precision magnetic ranging technology in the form of a detection array used in this paper can be applied in the currently popular field of robotics, where array-based magnetic ranging technology is utilized for robot navigation and positioning. In the engineering field, this technology can be used to measure the adjacent distances in paired drilling operations within drilling projects. In the medical device field, it can be employed to obtain the position of wireless capsule endoscopes and the orientation of the capsule within the human body. The cable conductor positioning analysis algorithm based on the cosine theorem and Pythagorean theorem presented in this paper can further extend its utility to fields such as robot motion posture analysis. This demonstrates that the magnetic field-based cable eccentricity detection proposed in this paper not only has multi-domain applications but also enhances the commercial value of these application entities, thereby confirming that the described eccentricity detection method possesses high versatility, practicality, and commercial potential.The specific location in the paper is line 574-584, page 18, highlighted in yellow for emphasis.

Comments 6:Some phrases could be simplified to avoid redundancy. For example, "the magnetic flux density cross-sectional simulation diagram as depicted in Figure 2(a)" could be rephrased as "the magnetic flux density cross-sectional simulation diagram in Figure 2(a)"。

Response 6:Thanks for your valuable English comment. We agree with this comment. We have invited nativea In streamlining verbose phrases, we sought advice from the teachers and students in our university's English department. With their assistance, we rectified the issue of verbosity in the phrases within the paper.Thank you again for the comment, it has been very helpful to us.

Comments 7:Most of the document uses the present tense consistently, which is appropriate for describing a method. However, occasionally the past tense is used in describing the method (e.g., "As depicted in the detection depicted in the detection cross-sectional diagram..."). Ensuring consistency in tense throughout the document would make it more cohesive.

Response 7:Thanks for your valuable English comment. We agree with this comment. In the process of standardizing the tense to the present tense throughout the paper, we consulted with professors and students from the English Department of our university. With their assistance, we have corrected the inconsistencies in tenses within the paper.Thank you again for the comment, it has been very helpful to us.

  Special thanks to you for your good comments.We tried our best to improve the manuscript and made some changes in the manuscript. These changes will not influence the content and framework of the paper.

  We appreciate for your warm work earnestly, and hope that the correction will meet with approval. Once again, thank you very much for your comments and suggestions.

Best regards.

Yours sincerely,

Yuandi Liu,Pengxuan Wei,Yinghong Zhang

E-mail: 13328280121@163.com

Tel: +86 13328280121

Address: Guilin University of Electronic Technology, Huajiang Campus, Lingtian Town, Lingchuan County, Guilin City, Guangxi Zhuang Autonomous Region.

Reviewer 2 Report

Comments and Suggestions for Authors

The authors studied the cable eccentricity detection using magnetic field. This is a good try while I have the following comments of this paper:

(1) Regarding the title, I think “Cable Eccentricity Detection Method Based on Magnetic Field” would be sufficient. 

(2) The Abstract is too long. Please constrain it within 200 words.

(3) Please specify what kind of cables (e.g., voltage, one or three cores, etc.) the GB/T 12706-2020 and IEC6502.2 refer to. Otherwise the readers cannot understand why the GB/T is more loose than IEC60502.2.

(4) P. 2: you stated that “these methods are characterized by several disadvantages, including high detection costs, poor accuracy, low efficiency, and inadequate stability.” However, after reading this paragraph, I still cannot understand how it is. Please quantify your statement. 

(5) PP. 2 – 3: you listed out so many techniques on cable eccentricity detection from the world. I would suggest you to use a table for a comparison, otherwise it looks so lengthy. By the way, you didn’t state out your purpose of writing this. Do you say these techniques are inadequate?

(6) 2.1, Principle: is your method only applicable to one-core un-shielded cable?

(7) PP. 7-8, Calculation: you should give the final equation regarding the relationship between magnetic fields of your six sensors, and the cable eccentricity. This is important for your technology. 

(8) Experiment: you said that the current intensity and lift-off value are two important factors for your method. Please specify how to control the two parameters regarding the test for different cable rating and physical appearance. 

(9)  Experiment: how accurate would be the requirement for your magnetic sensors? For example, if the measurement is affected by external magnetic fields, or the error of one sensor measurement, how would it affect the cable eccentricity calculation? 

(10) Reference: I think you have 20-40 references would be enough. The no. of 77 is a bit astonishing. 

I suggest the authors to contemplate them seriously for a review again; otherwise I won’t support its publication as a reviewer.  

Comments on the Quality of English Language

It's OK.

Author Response

August 9, 2024

Dear Editors and Reviewer,

  Thank you for your letter and for the reviewers’ comments concerning our manuscript entitled “Cable Eccentricity Detection Method Based on Magnetic Field” (ID: sensors-3128793). Those comments are all valuable and very helpful for revising and improving our paper, as well as the important guiding significance to our researches. We have studied comments carefully and have made correction which we hope meet with approval. Revised portion are marked with color of yellow in the paper. The main corrections in the paper and the responds to the reviewer’s comments are as flowing:

Comments 1:Regarding the title, I think “Cable Eccentricity Detection Method Based on Magnetic Field” would be sufficient. 

Response 1:Thank you for your valuable insights. We agree with this comment. We have changed the paper title to: “Cable Eccentricity Detection Method Based on Magnetic Field”. The specific location in the paper is line 2, page 1, highlighted in yellow for emphasis.

Comments 2:The Abstract is too long. Please constrain it within 200 words.

Response 2:Thank you for your valuable insights. We agree with this comment. We have refined the abstract to under 200 words while ensuring the integrity of the content remains intact. The specific location in the paper is line 7-20, page 1, highlighted in yellow for emphasis.

Comments 3:Please specify what kind of cables (e.g., voltage, one or three cores, etc.) the GB/T 12706-2020 and IEC6502.2 refer to. Otherwise the readers cannot understand why the GB/T is more loose than IEC60502.2.

Response 3:Thank you for your valuable insights. We agree with this comment. Through a meticulous review of the requirements for cable eccentricity in GB/T 12706-2020 and IEC 60502.2, it has been discovered that our prior conclusion suggesting the requirements for cable eccentricity in IEC 60502.2 are more stringent than those in GB/T 12706-2020 was incorrect. The accurate conclusion is that for all cable voltage rating ranges and materials, the requirement for eccentricity is that it should not exceed 15%. We extend our apologies for the erroneous conclusion stated in the previous manuscript. The specific location in the paper is line 27-33, page 1, highlighted in yellow for emphasis.

Comments 4:P. 2: you stated that “these methods are characterized by several disadvantages, including high detection costs, poor accuracy, low efficiency, and inadequate stability.” However, after reading this paragraph, I still cannot understand how it is. Please quantify your statement. 

Response 4:Thank you for your valuable insights. We agree with this comment. In this version of the manuscript, we have provided a detailed quantitative description of the shortcomings of traditional detection methods such as X-ray inspection, eddy current testing, and optomagnetic detection. We have listed specific data, for example, the equipment cost for X-ray detection ranges from $10,000 to $200,000, and the detection speed is 0.5-5 meters per second, clarifying the specific causes of the defects associated with these three methods.The specific location in the paper is line 45-61, page 2, highlighted in yellow for emphasis.

Comments 5:PP. 2 – 3: you listed out so many techniques on cable eccentricity detection from the world. I would suggest you to use a table for a comparison, otherwise it looks so lengthy. By the way, you didn’t state out your purpose of writing this. Do you say these techniques are inadequate?

Response 5:Thank you for your valuable insights. We agree with this comment. In this version of the manuscript, we have compiled the analyzed literature into a table for supplementary explanation, facilitating readers to quickly review and compare the differences and respective shortcomings between various magnetic ranging and positioning methods. Additionally, we have clarified the purposes of analyzing these papers in the paper:

1) To illustrate that the current magnetic ranging and positioning methods have reference significance for the cable eccentricity detection method proposed in this paper, possessing the potential to be applied to the precise measurement of small distances such as cable eccentricity detection, thereby substantiating the feasibility of the magnetic field-based cable eccentricity detection method presented in this paper.

2) To indicate that these technologies are primarily applied in larger or large-scale engineering scenarios, such as drilling projects, where the errors and measurements are too significant for cable eccentricity detection, exhibiting limitations in precise small-distance measurement. Therefore, the innovation of this paper lies in applying magnetic ranging technology to the precise small-distance measurement scenario of cable eccentricity detection. The purpose of the literature review within the paper is located on line 127-135.

The specific location in the paper is line 127-135, page 3、line159, page 4-5, highlighted in yellow for emphasis.

Comments 6:2.1, Principle: is your method only applicable to one-core un-shielded cable?

Response 6:Thank you for your valuable insights. We agree with this comment. The cable eccentricity detection method is capable of detecting the eccentricity of unshielded single/multi-core cables by measuring the overall eccentricity of the cable. The detection method for single-core cables is as described in the paper. When applying the detection method to multi-core cables, the multiple small cores intertwined within the cable are regarded as a single large conductor, and the same method used for detecting the eccentricity of single-core cables is then employed to measure the overall cable eccentricity, which is considered the eccentricity of the multi-core cable. However, due to the characteristic of the proposed method that the magnetic field is generated internally by the cable and detected externally, this method is not applicable for cables with a metallic shield. Therefore, the magnetic field-based cable eccentricity detection method presented in this paper can measure the eccentricity of unshielded single/multi-core cables without a metallic shield.The specific location in the paper is line 162-173, page 5, highlighted in yellow for emphasis.

Comments 7:PP. 7-8, Calculation: you should give the final equation regarding the relationship between magnetic fields of your six sensors, and the cable eccentricity. This is important for your technology. 

Response 7:Thank you for your valuable insights. We agree with this comment. In this version of the manuscript, we present the transformation of the magnetic flux density values B, as detected by the magnetic field acquisition module, to the final cable eccentricity e calculations in the form of a general equation set, displayed as Equation 8 within the paper. In practical detection, the detected magnetic flux density values B are substituted into this set of equations. Utilizing the cosine theorem and the cosine sum theorem, in conjunction with the MCU's numerical solution library for numerical computation, we determine the coordinates of the cable's conductor center within the detection coordinate system. Subsequently, we calculate the eccentricity distance △E, compare the relationship between △E and the conductor's radius r, and calculate the maximum and minimum thicknesses t_max and t_min of the cable's protective jacket. Based on t_max and t_min, the final cable eccentricity e is computed. The specific location in the paper is line 344-380, page 10-12, highlighted in yellow for emphasis.

Comments 8:Experiment: you said that the current intensity and lift-off value are two important factors for your method. Please specify how to control the two parameters regarding the test for different cable rating and physical appearance. 

Response 8:Thank you for your valuable insights. We agree with this comment. By increasing the current intensity and reducing the lift-off value, we can enhance the magnetic flux density around the magnetic field collection array. This leads to more precise and reliable measurements of cable eccentricity, ensuring better stability across a series of repetitive tests. In practice, the current applied to the cable is usually set between half and the full rated current, and the lift-off value, measured from the cable surface, is generally set between 1-4mm for optimal results. In actual detection, we first determine the lift-off value and then adjust the applied current based on the detection effects to achieve the desired outcome. For example, in the case of a polyvinyl chloride (PVC/A) cable with a rated voltage of 0.6/1 (1.2) kV, a typical rated current of 100A, and a conductor cross-sectional area of 25 square millimeters, a lift-off value of 1-5mm is generally chosen, and a current ranging from 50A to 100A is set as the test loading current for the detection of cable eccentricity. The specific location in the paper is line 512-528, page 17, highlighted in yellow for emphasis.

Comments 9:Experiment: how accurate would be the requirement for your magnetic sensors? For example, if the measurement is affected by external magnetic fields, or the error of one sensor measurement, how would it affect the cable eccentricity calculation? 

Response 9:Thank you for your valuable insights. In the event of an error occurring in one of the sensors within the magnetic field acquisition array, the cable eccentricity detection system that we have designed will compare the measurement data obtained from the sensor's position and its opposing position in the array. If there is a significant deviation from the cable's cross-sectional parameters, the system will proactively discard the data set from that particular pair of opposing positions. For example, if the sum of the conductor's distance to the left sheath closing to  the left detection position (As shown at position "2'" in Figure 3(b).) and the right sheath closing to the right detection position(As shown at position "2" in Figure 3(b).) greatly exceeds the conductor's diameter plus the cable's specified cross-sectional dimensions, our cable eccentricity detection system will automatically reject this set of data. It will then assess the conformity of the data output from the remaining two pairs of positions with the standard cross-sectional parameters of the cable and select the data set with the best fit as the basis for the final eccentricity calculation.

when working in environments with significant electromagnetic interference, the cable eccentricity detection method described employs a preliminary sampling computation of the magnetic flux density values in the current environment before conducting live detection on the cable under test. This initial measurement establishes a baseline magnetic flux density value, referred to as the no-load value. Post-energization, a subsequent sampling computation is performed to ascertain the magnetic flux density once the cable is live, known as the loaded value. The actual magnetic flux density B induced by the energized cable is calculated by subtracting the no-load value from the loaded value. This derived value is then utilized to calculate the eccentricity of the cable, denoted as 'e'. This approach can effectively mitigate the impact of ambient electromagnetic disturbances on the cable eccentricity detection process.The specific location in the paper is line 529-558, page17-18, highlighted in yellow for emphasis.

Comments 10:Reference: I think you have 20-40 references would be enough. The no. of 77 is a bit astonishing. 

Response 10:Thank you for your valuable insights. We agree with this comment. In this edition of the manuscript, we have reduced the number of references to 38.The specific location in the paper is line 598-678, page 19-20, highlighted in yellow for emphasis.

Special thanks to you for your good comments.We tried our best to improve the manuscript and made some changes in the manuscript. These changes will not influence the content and framework of the paper.

  We appreciate for your warm work earnestly, and hope that the correction will meet with approval. Once again, thank you very much for your comments and suggestions.

Best regards.

Yours sincerely,

Yuandi Liu,Pengxuan Wei,Yinghong Zhang

E-mail: 13328280121@163.com

Tel: +86 13328280121

Address: Guilin University of Electronic Technology, Huajiang Campus, Lingtian Town, Lingchuan County, Guilin City, Guangxi Zhuang Autonomous Region.

Round 2

Reviewer 2 Report

Comments and Suggestions for Authors

I am fine with its publicatio now.

Comments on the Quality of English Language

I am fine with its publicatio now.